# Electrochemical Sensor Based on Laser-Induced Graphene for Carbendazim Detection in Water

**DOI:** 10.3390/foods12122277

**Published:** 2023-06-06

**Authors:** Li Wang, Mengyue Li, Bo Li, Min Wang, Hua Zhao, Fengnian Zhao

**Affiliations:** College of Chemistry and Materials Engineering, Beijing Technology and Business University, Beijing 100048, China

**Keywords:** carbendazim, laser-induced graphene, platinum nanoparticles, electrochemical detection, water sample

## Abstract

Carbendazim (CBZ) abuse can lead to pesticide residues, which may threaten the environment and human health. In this paper, a portable three-electrode sensor based on laser-induced graphene (LIG) was proposed for the electrochemical detection of CBZ. Compared with the traditional preparation method of graphene, LIG is prepared by exposing the polyimide film to a laser, which is easily produced and patterned. To enhance the sensitivity, platinum nanoparticles (PtNPs) were electrodeposited on the surface of LIG. Under optimal conditions, our prepared sensor (LIG/Pt) has a good linear relationship with CBZ concentration in the range of 1–40 μM, with a low detection limit of 0.67 μM. Further, the sensor shows good recovery rates for the detection of CBZ in wastewater, which provides a fast and reliable method for real-time analysis of CBZ residues in water samples.

## 1. Introduction

Carbendazim (CBZ) is a broad-spectrum fungicide with excellent activity and durability. It has been widely used in fruits and vegetables to protect crops from pathogenic bacteria and pests [1,2]. However, as a benzimidazole cyclic compound, carbendazim has strong stability and is not easily degraded in soil for a long time, thus causing serious residue problems and endangering environmental and food safety [3]. To date, CBZ has been banned in many countries, but it is still allowed in China [4]. The Ministry of Health and the Ministry of Agriculture have set maximum residue limits (MRLs) for CBZ in food to ensure that CBZ residues will not threaten human health. Therefore, it is of great significance to establish a rapid, sensitive, and effective method for the determination of CBZ residues for food safety [5].

At present, many methods have been developed to detect CBZ, including high-performance liquid chromatography (HPLC) [6], ultraviolet–visible spectrophotometry [7], capillary electrophoresis [8], fluorescence spectrometry [9], and electrochemical analysis [10]. Although sensitive and reliable, some methods have some drawbacks, such as complex sample pretreatment, high operation requirements, long detection time, and expensive instruments which are difficult to apply to rapid detection and on-site analysis of pesticide residues [11,12]. As a supplement, the electrochemical method is widely used in the detection of pesticide residues because of its low cost, high sensitivity, and simple operation [13,14]. At present, various electrochemical sensing methods have been used for the detection of CBZ [15]. To enhance analytical performance, carbon-based materials have been widely used in the field of electrochemical analysis due to their superior electrical properties [16]. Among them, graphene is a typical carbon-based material with a large surface area and favorable conductivity, which has been regarded as an excellent electrode material [17,18]. For example, Kumar et al. [19] used the nitrogen-doped reduced graphene oxide (RGO) as the nanocarrier to anchor gadolinium sesquisulfide for the electrochemical detection of CBZ in the river water sample. Further, Sundaresan et al. [20] constructed a highly sensitive electrochemical sensor using RGO as the functional platform to embed tungstate nanostructure for the detection of fenitrothion. Although these electrode materials have high sensitivity and good detection performance, the preparation process of graphene electrodes is complicated. To simplify the preparation of graphene electrodes, some advanced printing techniques were developed to broaden the application of graphene-based sensors in the analytical field [21,22,23].

Laser-induced graphene (LIG) is a porous graphene material with high electrocatalytic activity, a large surface area, and three-dimensional morphology [24]. The patterned LIG-based electrode can be easily obtained by scanning the surface of thermoplastic polymer materials (such as polyimide, PI) with the laser [25]. Previously, some low-cost, fast, sensitive, and flexible LIG-based electrochemical sensors were prepared on a polymer substrate by a laser direct engraving process [26]. However, single LIG devices usually exhibit limited performance sensitivity [27,28]. To solve this problem, various metal materials, especially platinum, silver, and gold nanoparticles (NPs), can be embedded in carbon carriers as connecting materials to improve sensitivity [29,30]. For example, Wang et al. [31] developed a label-free carcinoembryonic antigen (CEA) electrochemical immunosensor by anchoring AuNPs to LIG (LIG/Au) using chloroauric acid as a precursor. You et al. [30] fabricated laser-induced noble metal NPs (such as AuNPs, AgNPs, and PtNPs) and graphene composite, which were further applied to obtain the flexible electrode to realize the electrochemical detection of the pathogen.

Inspired by this, a strategy of detecting CBZ by the portable and integrated three-electrode electrochemical sensor based on LIG/Pt was put forward (Figure 1). Herein, three-dimensional porous LIG electrodes were prepared by laser direct writing on flexible PI thin films. To improve the sensitivity of the detection system, a PtNP-modified LIG sensor (LIG/Pt) was prepared by the electrodeposition method on porous LIG composites. Compared with the unmodified LIG electrode, the oxidation peak of CBZ on LIG/Pt was obviously enhanced, which proved that the prepared sensor had good electrochemical analysis performance. By connecting the prepared sensor with a handheld electrochemical workstation, the real-time information on CBZ residues was received on a smartphone, which could provide a simple, portable, and sensitive method for rapid and on-site analysis of CBZ in water samples.

## 2. Materials and Methods

### 2.1. Reagents and Instruments

CBZ, monocrotophos, methyl parathion, and phosphamidon were purchased from Shanghai Aladdin Bio-Chem Technology Co., Ltd. (Shanghai, China). Potassium chloroplatinite (K_2_PtC1_4_) was obtained from Shanghai Macklin Biochemical Co., Ltd. (Shanghai, China). Phosphate buffer solution (PBS, 0.1 M, pH 7.4) was purchased from Beijing Bairuiji Biotechnology Co., Ltd. (Beijing, China). The commercial PI film (thickness of 125 μm) and PI tape (80 μm) were purchased from DuPont, Wilmington, DE, USA. The Ag/AgCl paste was purchased from Ercon Inc., Wareham, MA, USA. All other chemicals and reagents used were of analytical grade.

The laser etching micromachine for preparing the LIG electrode was from Tianjin Jiayin Nanotechnology Co., Ltd. (Tianjin, China). All electrochemical measurements were carried out on a hand-held electrochemical workstation (EmStat3 Blue, PalmSens BV, Houten, The Netherlands) with a wireless Bluetooth transmission module. A field emission scanning electron microscope (HITACHI SU8010, Tokyo, Japan) was used for the morphology characterization of the prepared electrodes. A LabRAM HR Evolution Raman microscope system (Horiba Jobin Yvon, Kyoto, Japan) was operated to analyze the chemical composition of the porous graphene. An ESCALAB 250 X-ray photoelectron spectroscopy (ThermoFisher SCIENTIFIC, Waltham, MA, USA) was used to analyze the surface elemental composition of porous graphene. A 1200 liquid chromatograph (Agilent, Santa Clara, CA, USA) equipped with an AB5000 mass spectrometer (SCIEX, Boston, MA, USA) was taken as the gold standard.

### 2.2. Preparation of the LIG/Pt Sensor

In this experiment, the LIG electrode with porous structure was prepared by laser direct writing technique (Figure 1). The specific steps are as follows. The LIG-based three-electrode pattern was generated by using a computer to control a laser system to scan the surface of the PI film (power: 1.38 W; speed: 4.0 cm s^−1^). Then, the Ag/AgCl slurry was coated on the reference electrode and dried at 70 °C for 1 h. Finally, the electrode was encapsulated with the PI tape, thus realizing the preparation of the three-electrode LIG sensor.

The electrodeposition process of PtNPs was studied by the cyclic voltammetry (CV) method. 70 μL of 2 mM K_2_PtC1_4_ (solvent is 0.1 M Na_2_SO_4_) was dripped on the electrode area, and the parameters were set (scanning potential was from −0.4 V to +0.5 V, scanning rate was 50 mV s^−1^) with 20 cycles to prepare the LIG/Pt sensor.

### 2.3. Electrochemical Detection of CBZ

In this work, the square wave voltammetry (SWV) method was used to detect CBZ at room temperature, and all experiments were carried out on the sensor of a three-electrode LIG/Pt system combined with a hand-held electrochemical workstation. The electrochemical detection of CBZ mainly involves the following steps: firstly, clean the surface of LIG/Pt electrode with distilled water and dry it, and connect it with a hand-held electrochemical workstation. Then, a certain concentration of CBZ in 0.1 M PBS solution was dropped on the working area of the electrode. Finally, the SWV curve was recorded in the potential window of +0.4 to +1.0 V with the equilibrium equipment time of 15 s and the detection frequency of 10 Hz, respectively.

### 2.4. Real Sample Analysis

In this study, wastewater samples from a pig slaughterhouse in Hebei province were collected and used for real sample analysis. The water sample was diluted 5 fold (i.e., 1 mL of water sample was diluted with 4 mL 0.1 M PBS). Three groups of different spiking levels (10, 20, 30 μM) of CBZ were added to the water sample. Then, the mixture was vortexed and filtered through a 0.22 μm filter membrane. The supernatant was collected and used for the electrochemical analysis.

LC–MS/MS method was adopted to evaluate the accuracy. The analytical column used in chromatography is Waters XBridge C_18_ column, with 0.1% acetonitrile (A) and 0.1% formic acid solution (B) as the mobile phases, the column temperature was 40 °C, the flow rate was 200 μL min^−1^, and the injection volume was 10 μL, the gradient elution procedures are 80–60% B (0–2.0 min), 60–10% B (2.0–2.5 min), 10% B (2.5–3.0 min), 10–90% B (3.0–3.5 min) and 90% B (3.5–4.0 min). The mass spectrometry adopts an electrospray ionization (ESI) probe, the scanning mode was negative ion mode, and the temperature was 300 °C.

## 3. Results and Discussion

### 3.1. Characterization of the LIG Electrode

The preparation process of LIG is that a tightly focused laser beam can generate a high enough temperature on the target material to break its chemical bonds and rearrange carbon atoms into hexagonal graphene with characteristics [25,32]. In this study, the LIG-based three-electrode sensor was prepared by laser direct writing on the PI film as the substrate. During this process, the engraving power and scanning speed are the main influencing factors. Here, the engraving power of the laser is first characterized. When the power is too low (~1.1 W), the electrode pattern would not be completely printed. With the increase in laser power, the pattern on the PI film will be clearer. However, excessive power (~1.65 W) could damage the pattern and even puncture the PI film. Therefore, we first optimized the power with the same scanning speed (4.0 cm s^−1^). Three groups of LIG prepared with the most suitable engraving power were characterized by CV (Figure 2a) and differential pulse voltammetry (DPV, Appendix A) in 0.1 M KCl solution containing 1.0 mM K_3_[Fe(CN)_6_]. Results show that the electrochemical behavior of the LIG electrode is better at 1.38 W power. Then, the scanning speed was optimized under the optimal powder. Theoretically, the lower the scanning speed, the more graphene is formed. However, Figure 2b and Appendix A show that the LIG electrical activity at a scanning speed of 4.0 cm s^−1^ is higher than 2.4 cm s^−1^, which may be due to the collapse of the electrode structure at the slower scanning speed [33]. The size parameters of the LIG electrode are shown in Appendix A. The diameter of the working electrode area is 3.00 mm, and the length of the electrode sensing line is 5.25 mm. As shown in Appendix A, the LIG electrode prepared under the optimized conditions has a smooth silver-gray surface, and its size and morphology are consistent with the parameters involved.

After that, the morphology of the prepared LIG under the optimized conditions was characterized by SEM. As shown in Figure 2c and Appendix A, the fabricated LIG electrode appears typically porous structure, which is formed by the graphene sheets during laser scanning [27]. Raman spectroscopy and X-ray photoelectron spectroscopy (XPS) were used to characterize the chemical composition of the obtained materials. As shown in Figure 2d, three prominent peaks are observed in the Raman spectrum of LIG, namely, the D peak (~1350 cm^−1^), the G peak (~1580 cm^−1^), and the 2D peak (~2700 cm^−1^). Generally, the D peak reflects the lattice defects in graphene, while the G peak and the 2D peak reflect the characteristics of the prepared materials. By comparing the typical Raman spectra of graphene [34], it can be concluded that the materials we prepared belong to graphene materials. The C 1s spectrum of XPS analysis of LIG is shown in Appendix A. The main component of LIG, namely sp^2^-hybridized graphite carbon (C=C) [35], is presented at 284.5 eV, and some disordered carbon (C-C) is also observed at 285.5 eV. In addition, a small number of C-O-C groups were also observed at 287.5 eV.

### 3.2. Characterization of the LIG/Pt Sensor

Laser direct writing technology can be used to prepare LIG with controllable surface morphology, surface properties, chemical composition, and electrical properties, which can also minimize the use of raw materials. While noble NPs have the function of signal amplification, and thus embedding precious metal NPs on the surface of the LIG electrode will make it higher sensitivity and electrical activity [36]. In this paper, PtNPs were electrodeposited on the surface of LIG to form the LIG/Pt sensor. Here, several LIG/Pt sensors were prepared under different deposition cycles by the CV method. As shown in Figure 3a and Appendix A, the CV and DPV responses are increased after the modification of PtNPs, compared to those of bare LIG electrodes. When the electrodeposition cycles are 20, the response of the LIG/Pt sensor (labeled as LIG/Pt-20) reaches the most. As shown in Figure 3b and Appendix A, PtNPs (sizes of 70 ± 16 nm) are evenly distributed on the surface of LIG, which can increase the surface area of the electrode and further enhance the sensitivity.

After that, we investigated the electrochemical behavior of the obtained sensors for CBZ. Figure 3c shows the CV responses of the LIG/Pt sensor and bare LIG sensor with or without 10 μΜ CBZ. It is worth noticing that the LIG/Pt sensor has a higher response current than the bare electrode, which is because PtNPs can accelerate the electron transfer rate and amplify the background signal of LIG. After dropping 10 μM CBZ on the electrode surface, an obvious oxidation peak (O peak) appears at +0.77 V on the CV curve, which indicates that CBZ is oxidized to methyl carbamate and corresponding benzimidazole radical on the LIG/Pt sensor by the CV method [37]. The oxidation process of CBZ involves four electrons, and the possible oxidation mechanism is shown in Figure 3d. In contrast, the bare LIG sensor shows a low peak response due to its slow electron transfer rate. According to the above results, it can be seen that our prepared LIG/Pt sensor shows a favorable potential in the detection of CBZ.

### 3.3. Optimization of Experimental Parameters

Although the sensitivity of the SWV method may be lower than that of the DPV method, the faster analytical speed makes it unique in the rapid and on-site analysis. Therefore, the determination performance of the LIG/Pt sensor was studied via the SWV method in this experiment. Herein, the equilibrium time and scanning frequency were optimized to obtain the optimal instrumental parameters. Firstly, we set various equilibrium time (0 s, 10 s, 15 s, 20 s, 25 s) with the same frequency of 10 Hz. The original data and calibrated data of SWV are shown in Figure 4a,d. Results indicate that the current response of SWV for 10 μM CBZ is better when the equilibrium time is 15 s. After that, the scanning frequency was optimized with the optimal equilibrium time of 15 s. Figure 4b,e show the original data and calibrated data of SWV response signals at different frequencies. It is clear that the current response increases with the increment of scanning frequency. When the scanning frequency reaches 10 Hz, the current response increases to a maximum. Thus, we choose 10 Hz as the optimal frequency for CBZ detection.

Finally, the SWV response of CBZ was optimized in various pH values of the working solution (0.1 M PBS). As shown in Figure 4c, with the increase in pH, the peak potential shifts negatively, which indicates that protons may participate in the reaction process at the modified electrode. In addition, by observing the calibrated image (Figure 4f), it can be seen that with the increase in pH value, the peak current first increases and then decreases. Therefore, the optimal pH value for the working solution was 7.

### 3.4. Electrochemical Detection of CBZ

Under the optimized conditions, the corresponding relationship between the CBZ concentration and SWV current signal was investigated. As shown in Figure 5a, the peak current response gradually increases with the increase in CBZ concentration. To obtain the current peak more conveniently, Origin software was used to correct the curves. Based on the baseline-corrected data in Figure 5b, a calibration curve is fitted with CBZ concentration as abscissa and current response as ordinate (Figure 5c). Results indicate that when the concentration of CBZ is in the range of 1−40 μM, the corresponding current signal has a linear relationship with its concentration, the regression equation is y = 2.0142x + 0.0022, and the correlation coefficient R^2^ is 0.9998. According to the formula of detection limit (LOD) = 3 Sb/m (Sb is the standard deviation of SWV in the blank experiment and m is the slope of the calibration curve), the LOD is 0.67 μM (i.e., 128.1 μg L^−1^), which is far below than the MRLs stated by China in many food samples, such as some vegetables and fruits.

In addition, the performance of our sensor was also compared with other reported electrochemical sensors [11,37,38,39]. As shown in Table 1, our prepared LIG/Pt sensor has comparable analytical performance for the detection of CBZ.

### 3.5. Anti-Interference, Selectivity, Reproducibility, and Stability Analysis

In the real sample analysis, some non-target components may interfere with the detection process due to the complexity of the sample matrix. Firstly, we investigated the anti-interference ability of the LIG/Pt sensor. Several possible ions (Ca^2+^, K^+^, Na^+^, 100-fold higher than CBZ) and small molecules (glucose and ascorbic acid, 50-fold higher than CBZ) that may coexist or exist during the real sample analysis were selected in this study. As shown in Appendix A, there is no obvious difference in electrical signals in the experiment which indicates the favorable anti-interference performance of our LIG/Pt sensor.

Moreover, we evaluated the selectivity of our sensor. Herein, the SWV signals of the LIG/Pt were recorded in the same concentration (10 μM) of CBZ, monocrotophos, methyl parathion, and phosphamidon, respectively. As shown in Appendix A, the LIG/Pt only exhibits a clear SWV signal for CBZ, indicating the high selectivity of our sensor for CBZ.

Reproducibility and stability are important performance parameters of the electrode. Five electrodes were prepared under the same conditions, and the reproducibility of the LIG/Pt electrode was evaluated by detecting 10 μM CBZ with the SWV method. As shown in Appendix A, the current response ratio of five electrodes has little difference, with the RSD not over 1.79%, indicating the favorable repeatability of our sensor. Further, the stability of the LIG/Pt sensor was assessed by storing it for several days at room temperature. Results show that the current response ratios in five days are from 94.8% to 103.8%, with the RSD below 3.26%, declaring the satisfying storage stability of our sensor (Appendix A).

### 3.6. Real Sample Analysis

In this paper, wastewater was selected as the real sample, and the recovery experiment was adopted for methodological evaluation. We set three spiking levels (10, 20, 30 μM) in blank water samples. After placing for 0.5 h at room temperature, the above samples were diluted 5 fold with 0.1 M PBS (pH = 7) and then filtered through a 0.22 μm membrane. The supernatant was collected for detection. As shown in Table 2, the recoveries of CBZ in wastewater range from 88.89% to 99.50%, with RSDs of 1.98–5.11% (*n* = 3). To evaluate the accuracy of the sensor detection, the CBZ concentration was measured simultaneously by the LC–MS/MS standard method. The results show that the determination of CBZ in water samples by LIG/Pt is similar to that by LC–MS/MS, indicating the satisfying accuracy of our prepared LIG/Pt sensor. In the future, our efforts will be directed towards applying this sensor in more complicated food samples by the improved sample pretreatment method, such as the adoption of efficient cleaning agents.

## 4. Conclusions

In this study, a portable LIG-based electrochemical sensor was developed for the rapid detection of CBZ in water samples. By laser direct writing on the PI film, the LIG three-electrode sensor was easily obtained. The structure and composition of LIG were verified by SEM, Raman spectroscopy, and XPS, which confirmed its highly porous graphene structure and excellent specific surface area. By electrodepositing PtNPs on the surface of LIG, the detection performance was further enhanced. Under optimal conditions, the prepared sensor (LIG/Pt) has a wide linear range (1–40 µM), a satisfactory LOD (0.67 µM), and good recoveries (88.89–99.50%) in wastewater samples. The electrochemical sensor prepared in this study is simple in operation, high in sensitivity, and good in selectivity, which can provide a reliable and real-time analysis method for the detection of CBZ residues in water samples.

## Figures and Tables

**Figure 1 foods-12-02277-f001:**
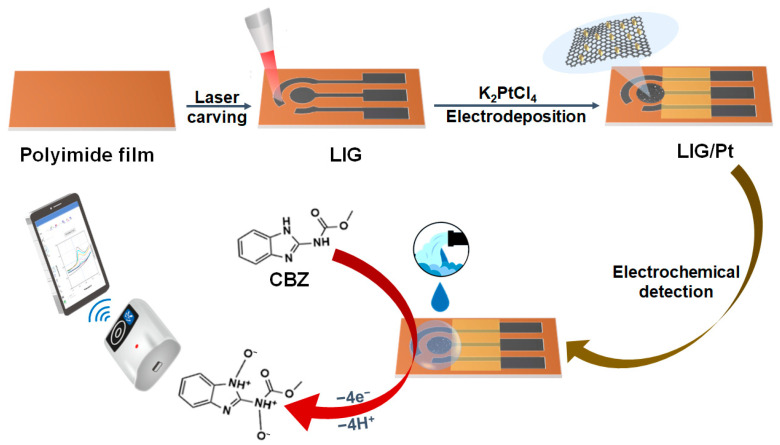
Preparation of LIG/Pt electrochemical sensor and its application for the real-time detection of CBZ in wastewater samples.

**Figure 2 foods-12-02277-f002:**
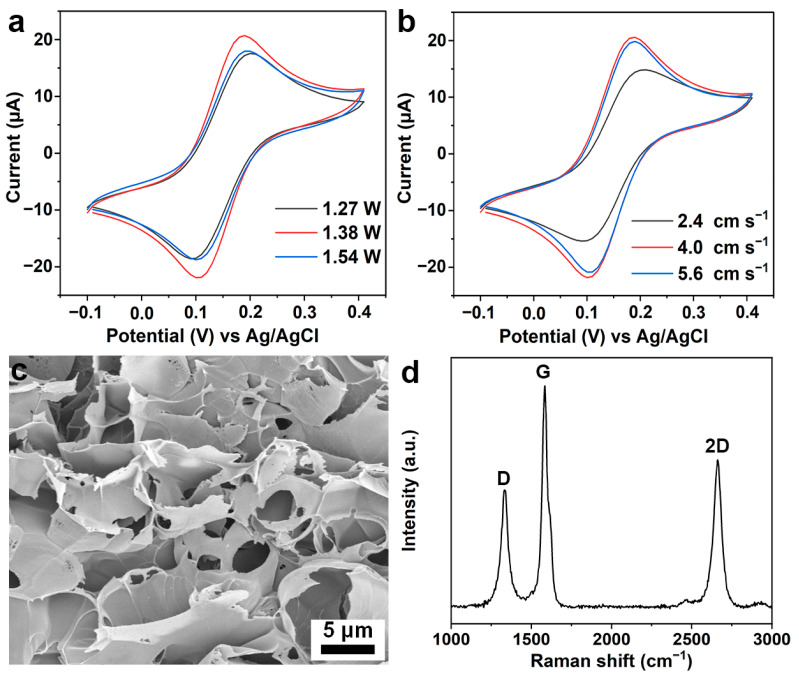
Characterization of the prepared LIG electrode. CV curves of the prepared electrodes with various (**a**) power and (**b**) scanning speed parameters in 0.1 M KCl solution containing 1.0 mM K_3_[Fe(CN)_6_]. (**c**) SEM image and (**d**) Raman spectrum of the prepared LIG electrode under optimal conditions (1.38 W and 4.0 cm s^−1^).

**Figure 3 foods-12-02277-f003:**
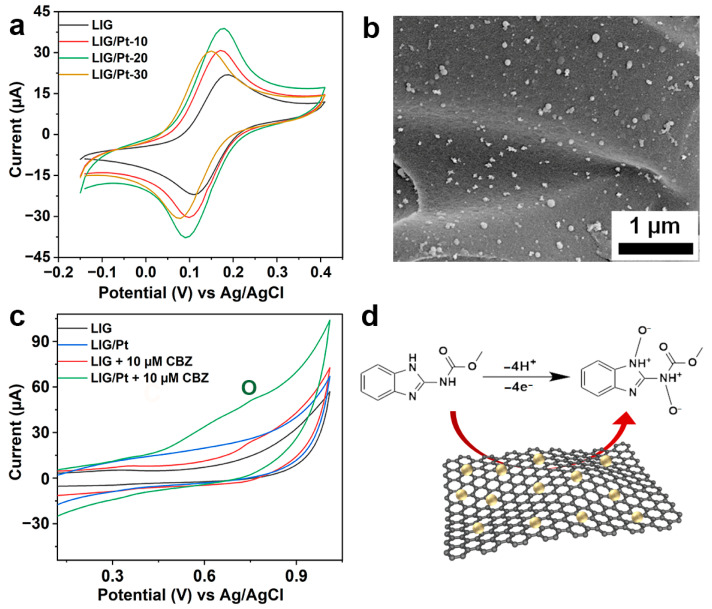
Characterization of the prepared LIG/Pt sensor. (**a**) Optimization of the electrodeposition cycles of the LIG/Pt sensor. (**b**) SEM image of the LIG/Pt sensor under optimal conditions. (**c**) CV responses of the LIG/Pt and bare LIG with or without 10 μΜ CBZ. (**d**) The electrochemical detection mechanism of CBZ.

**Figure 4 foods-12-02277-f004:**
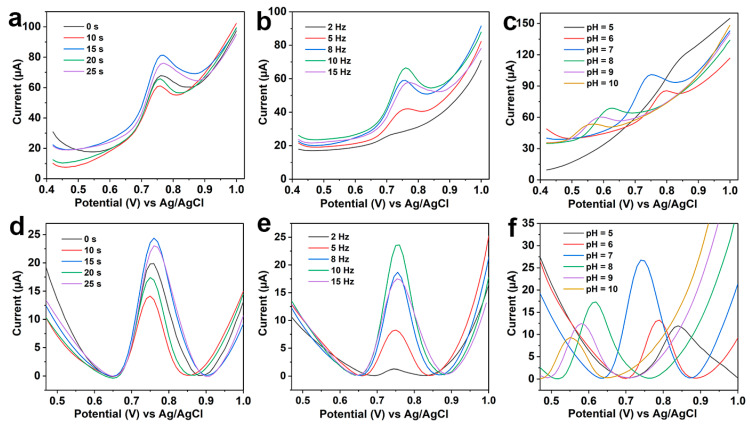
Parameter optimization of SWV responses of the LIG/Pt sensor for CBZ. The original SWV response data with various (**a**) equilibrium times, (**b**) detection frequencies, and (**c**) pH values of 0.1 M PBS solution. (**d**–**f**) The corresponding baseline-corrected data.

**Figure 5 foods-12-02277-f005:**
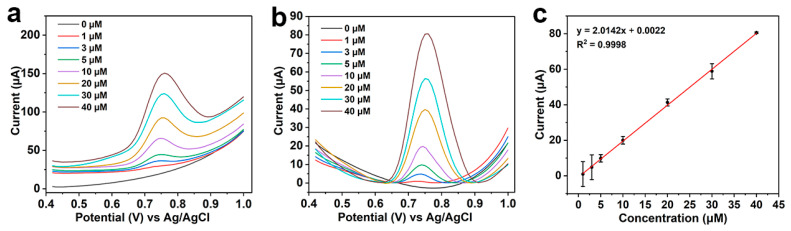
SWV responses of the LIG/Pt sensor for CBZ with the concentration in the range of 0–40 μM. (**a**) The SWV response data initial image. (**b**) The SWV response data with baseline correction. (**c**) The corresponding calibration curve between CBZ concentration (1–40 μM) and SWV current signal.

**Table 1 foods-12-02277-t001:** Comparison of CBZ determination by electrochemical sensors with different electrodes.

Sensor	Method	Linear Range	LOD	Ref.
GCE/PPy-CNT	DPV	4–20 μM	1.3 μM	[39]
GCE/MBC@CTS	DPV	0.1–20 µM	0.02 μM	[11]
GCE/RGO-Pt	DPV	25–115 μM	2.96 μM	[38]
GCE/RGO/NP-Cu	DPV	0.5–30 μM	0.09 μM	[37]
GCE/NiCo-LDH	DPV	0.006–14.1 μM	0.001 μM	[40]
LIG/Pt	SWV	1–40 μM	0.67 μM	This work

Notes: GCE, Glassy carbon electrode; PPy, polypyrrole; CNT, carbon nanotube; MBC@CTS, mung bean-derived porous carbon@chitosan; RGO, reduced graphene oxide; NP-Cu, nanoporous copper; NiCo-LDH, nickel cobalt-layered double hydroxide.

**Table 2 foods-12-02277-t002:** Recovery study of CBZ in wastewater samples (*n* = 3).

Method	Added (µM)	Founded (µM)	Recovery (%)	RSD (%)
This method	10	9.95 ± 0.20	99.50	1.98
	20	17.78 ± 1.61	88.92	4.12
	30	26.67 ± 1.10	88.89	5.11
LC–MS/MS	10	8.51 ± 0.11	85.14	1.27
	20	17.44 ± 0.55	87.22	3.14
	30	27.07 ± 1.69	90.24	6.23

## Data Availability

The data used to support the findings of this study can be made available by the corresponding author upon request.

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
