# Peer review of "Electrochemical Sensor Based on Laser-Induced Graphene for Carbendazim Detection in Water"

_foods, 2023, doi:10.3390/foods12122277_

Round 1
Reviewer 1 Report
foods-2401234
Review of the article titled:
«Portable Electrochemical Platform Based on Laser-induced Graphene for Carbendazim Detection»
by Li Wang, Mengyue Li, Bo Li, Min Wang, Hua Zhao, Fengnian Zhao
In Foods (ISSN 2304-8158).
Round 1
The abstract perfectly underlines the relevance of the presented research. However, despite many proof-of-concept demonstrations, the full potential of carbendazim analysis has yet to be investigated.
The article was presented in a well-structured manner, with a good level of organization. Unfortunately, several statements within have weak evidence. Therefore, the referee suggested improving the manuscript with a major revision. The following is a list of specific concerns.
1. I need help understanding the procedure of laser-induced graphene preparation. Is it pristine graphene without functional groups? It is highly recommended to present XPS data to prove the surface composition. Also, provide the schematic procedure of LIG preparation, including the preparation condition.
2. Provide information concerning the sensor's stability and reproducibility (RSD,%).
3. Provide real sample spiked analysis (e.g. wastewater) to evaluate the matrix influences on results.
4. The title of the article should be specified.
5. The novelty of the article could be more straightforward.
6. Please specify the axis description as follows. Replace Potential (V) to Potential (V) vs Ag|AgCl (to figures 2a,b; 3a,b; 4; 5a,b.
Author Response
Reviewer 1.
The abstract perfectly underlines the relevance of the presented research. However, despite many proof-of-concept demonstrations, the full potential of carbendazim analysis has yet to be investigated. The article was presented in a well-structured manner, with a good level of organization. Unfortunately, several statements within have weak evidence. Therefore, the referee suggested improving the manuscript with a major revision. The following is a list of specific concerns.
Response: Thank you so much for your positive comments. We have tried our best to revise our manuscript carefully and hope that the correction will meet with your approval.
Question 1. I need help understanding the procedure of laser-induced graphene preparation. Is it pristine graphene without functional groups? It is highly recommended to present XPS data to prove the surface composition. Also, provide the schematic procedure of LIG preparation, including the preparation condition.
Response: Thank you very much for your advice. We have added XPS analysis in the supplementary document. From the C 1s spectrum of XPS analysis of LIG in Figure S3, it can be seen that there are a large number of sp2 hybrid graphite carbons (C=C), some disordered carbons (C-C), and a small number of C-O-C groups in LIG. Furthermore, the detailed electrode preparation process is added to Figure 1.
Question 2. Provide information concerning the sensor's stability and reproducibility (RSD, %).
Response: Thank you very much for your advice. We have added the content of reproducibility and stability in Figure S5c-d. Five electrodes were evaluated by detecting 10 μM CBZ with the SWV method for the reproducibility test. As shown in Figure S5c, the current response ratio of five electrodes has little difference, with the RSD not over 1.79%, indicating the favorable repeatability of our sensor. Besides, the stability of the LIG/Pt sensor was assessed by storing it for several days at room temperature. Results show that the current response ratios in five days are between 94.8% to 103.8%, with the RSD less than 3.26%, declaring the satisfying storage stability of our sensor (Figure S5d).
Question 3. Provide real sample spiked analysis (e.g. wastewater) to evaluate the matrix influences on results.
Response: Thank you very much for your advice. We collected the wastewater from the local pig slaughterhouse and investigated the recovery experiment. The corresponding data were added in the revised Table 2. To evaluate the accuracy of the sensor detection, the CBZ concentration was also measured by LC-MS/MS standard method.
Question 4. The title of the article should be specified.
Response: Thank you very much for your advice. We have revised the title to make it clearer.
Question 5. The novelty of the article could be more straightforward.
Response: Thank you very much for your advice. We intuitively described the innovation of this paper in the section ‘Abstract’ and ‘Conclusion’.
Question 6. Please specify the axis description as follows. Replace Potential (V) to Potential (V) vs Ag/AgCl (to figures 2a,b; 3a,b; 4; 5a,b.)
Response: Thank you very much for your advice. We have corrected the relevant pictures in the revised manuscript.
Reviewer 2 Report
The manuscript entitled " Portable electrochemical platform based on laser-induced graphene for carbendazim detection" reports the synthesis of laser-induced graphene and further utilization as the electrode material for carbendazim detection. In my opinion, the manuscript is well written and organized. Although the idea is great and novel, it needs some further experiments to improve the quality of the manuscript. Moreover, the following comments should be considered
Comments:
1. The synthesis scheme is not clear, and it should be revised in a detailed way.
2. The authors need to move the comparison table to main text and included the following articles:
https://doi.org/10.1016/j.surfin.2022.102570
3. Please calculate the accuracy of the method by comparing the method with other standard method.
4. Authors should be trimmed/condense the ‘Abstract’ and ‘Conclusion’ sections in the revised manuscript. Please keep highlights of the whole manuscript in both sections.
5. This manuscript has some spelling typos, style errors, and grammatical errors, which severely affect its readability. So, I suggest the authors carefully check the whole manuscript and correct them.
Author Response
The manuscript entitled "Portable electrochemical platform based on laser-induced graphene for carbendazim detection" reports the synthesis of laser-induced graphene and further utilization as the electrode material for carbendazim detection. In my opinion, the manuscript is well written and organized. Although the idea is great and novel, it needs some further experiments to improve the quality of the manuscript. Moreover, the following comments should be considered.
Response: Thank you so much for your positive comments. We have tried our best to revise our manuscript carefully according to your suggestion.
Question 1. The synthesis scheme is not clear, and it should be revised in a detailed way.
Response: Thank you very much for your advice. We have revised the scheme and supplemented the specific synthesis flow chart in Figure 1.
Question 2. The authors need to move the comparison table to main text and included the following articles: https://doi.org/10.1016/j.surfin.2022.102570
Response: Thank you very much for your advice. We have moved the comparison table to the main text and added the comparison with the above article in the revised Table 1.
Question 3. Please calculate the accuracy of the method by comparing the method with other standard method.
Response: Thank you very much for your advice. In the revised manuscript, the CBZ concentrations in wastewater samples were measured simultaneously by LC-MS/MS. To evaluate the accuracy of the sensor detection, the CBZ concentration was measured simultaneously by LC-MS/MS standard method. The results show that the determination of CBZ in water samples by LIG/Pt is similar to that by LC-MS/MS, indicating the satisfying accuracy of our prepared LIG/Pt sensor.
Question 4. Authors should be trimmed/condense the ‘Abstract’ and ‘Conclusion’ sections in the revised manuscript. Please keep highlights of the whole manuscript in both sections.
Response: Thank you very much for your advice. The ‘Abstract’ and ‘Conclusion’ sections have been streamlined in the revised manuscript.
Question 5. This manuscript has some spelling typos, style errors, and grammatical errors, which severely affect its readability. So, I suggest the authors carefully check the whole manuscript and correct them.
Response: Thanks so much for your advice. The whole manuscript has been checked and revised carefully.
Reviewer 3 Report
The main issues of the ms are:
1. the title is misleading, it should be noted that the method was developed for tap water;
2. a DoE have to be performed when an optimization for a new method is proposed. Optimizing variables in that way does not involve the evaluation of their interaction. In addition, if the optimal way was on an intermediary stadium, it is impossible to detect.
3. a comparison whit established methods for analyte detection or at least with the golden standards (also from literature) in terms of method figure of merits must be provided
Author Response
Question 1. The title is misleading, it should be noted that the method was developed for tap water.
Response: Thanks for your comments, we have revised the title to make it clear according to the content of the manuscript.
Question 2. A DoE have to be performed when an optimization for a new method is proposed. Optimizing variables in that way does not involve the evaluation of their interaction. In addition, if the optimal way was on an intermediary stadium, it is impossible to detect.
Response: Thank you for your very instructive advice. We agree that DoE, a systematic and logical method, is helpful to optimize the experimental parameters. During our study, we mainly referred to the following literature to design the optimization experiment (e.g., ACS Applied Nano Materials 2022, 5, (12), 17516-17525; Biosensors and Bioelectronics 2020, 170, 112636), which can realize the sensitive detection of CBZ in water samples. We believe that better analytical performance can be obtained via the DoE. We will seriously consider your suggestion and conduct further research in the future, hoping to use it in the parameter optimization of the new method in the future.
Question 3. A comparison whit established methods for analyte detection or at least with the golden standards (also from literature) in terms of method figure of merits must be provided.
Response: Thank you for your valuable comments. In the revised manuscript, the more complicated water samples (i.e., wastewater samples) were tested for recovery study. CBZ concentrations in wastewater samples were measured simultaneously by LC-MS/MS. The results show that the determination of CBZ in water samples by LIG/Pt is similar to that by LC-MS/MS, indicating the satisfying accuracy of our prepared LIG/Pt sensor.
Round 2
Reviewer 1 Report
The authors addressed my comments well. I would recommend its acceptance.
Reviewer 2 Report
The authors answered all questions and completed the changes. The manuscript in its current form is acceptable.
Reviewer 3 Report
Accepted